# Welfare Assessment, End-Point Refinement and the Effects of Non-Aversive Handling in C57BL/6 Mice with Lewis Lung Cancer

**DOI:** 10.3390/ani12010023

**Published:** 2021-12-23

**Authors:** Amy L. Miller, Johnny V. Roughan

**Affiliations:** 1School of Natural and Environmental Sciences, Newcastle University, Newcastle upon Tyne NE1 7RU, UK; Amy.Miller@newcastle.ac.uk; 2Institute of Bioscience, Newcastle University, Newcastle upon Tyne NE2 4HH, UK

**Keywords:** mouse welfare, lung cancer, pain, nest building, IVIS, behaviour

## Abstract

**Simple Summary:**

Mice are widely used to study various types of cancer to develop effective treatments, but this can cause them to experience pain or poor welfare, which may lead to inconsistent or negative findings. This study used a variety of welfare monitoring methods to determine if, or more importantly when mice developing lung cancer experienced poor welfare so that similar future studies can be ended before this occurs. It also aimed to determine whether non-aversive handling, also known as tunnel handling (lifting mice in a plastic tube rather than by the tail) is an effective method of refinement, leading to improved welfare and improved consistency of results. Compared to normal mice, those developing lung cancer showed behaviour changes consistent with poor welfare including a general lack of movement and facial expression changes (grimacing). Most noticeably, cancer caused a reduction in food consumption and the ability of the mice to build a suitable nest. These combined factors suggested the mice began to suffer around 4 days before the study ended. Tunnel handling did not improve our results consistency or enhance welfare, but importantly, it had no negative effects. The findings suggest similar studies should be ended when mice appear to make reduced quality nests.

**Abstract:**

Cancer-bearing mice are at risk of developing anxiety, pain, or malaise. These conditions may not only harm welfare but could also undermine data quality and translational validity in studies to develop therapeutic interventions. We aimed to establish whether, or at what point mice developing lung cancer show these symptoms, what measures can best detect their onset, and if data quality and animal welfare can be enhanced by using non-aversive handling (NAH). Welfare was monitored using various daily methods. At the beginning and end of the study, we also scored behaviour for general welfare evaluation, recorded nociceptive thresholds, and applied the mouse grimace scale (MGS). Cancer caused a decline in daily welfare parameters (body weight, and food and water consumption) beginning at around 4 days prior to euthanasia. As cancer progressed, rearing and walking declined to a greater extent in cancer-bearing versus control mice, while grooming, inactive periods, and MGS scores increased. A decline in nest building capability and food consumption provided a particularly effective means of detecting deteriorating welfare. These changes suggested a welfare problem arose as cancer developed, so similar studies would benefit from refinement, with mice being removed from the study at least 4 days earlier. However, the problem of highly varied tumour growth made it difficult to determine this time-point accurately. There were no detectable beneficial effects of NAH on either data quality or in terms of enhanced welfare.

## 1. Introduction

Research animals experiencing pain, anxiety, or generally poor welfare may provide unreliable data; therefore, preventing unnecessary suffering is essential for scientific as well as ethical reasons. This may be particularly important in cancer-bearing mice where obtaining high quality data in relation to therapeutic interventions is fundamental to medical advancement. If mice suffer unnecessarily or beyond the limits that a particular study allows, then they may not yield suitably reliable findings, and the translational validity of those findings is likely to be undermined. The application of humane end-points in these circumstances is, therefore, a major opportunity to prevent unnecessary suffering, as well as enhance the accuracy and relevance of results. Although end-point guidelines are available for pre-clinical oncology studies (e.g., [1]) they do not offer model-specific advice. To try to address this issue we have undertaken several studies to determine whether commonly used cancer models cause pain to mice, and how severely. In those studies, we used standard welfare assessments including body weight and behaviour [2,3,4], but also used a novel application of the Conditioned Place Preference (CPP) procedure to determine that mice with orthotopically implanted bladder tumours were substantially more likely to experience pain than mice implanted heterotopically (i.e., subcutaneously) [2,3]. Those with orthotopically implanted tumours became listless, lost weight, developed hyperalgesia, and showed a progressive increase in conditioned morphine seeking. This suggested a potentially pain-related internal state alteration occurred as tumours developed [2]. As this was 7–10 days prior to the study ending, we recommended that the end-point of similar studies should be at least 1 week earlier. However, variable rates of cancer development between individual mice made it difficult to align tumour growth and spread with the time of onset of pain.

This study again aimed to establish the likely degree of suffering associated with cancer development, but in mice exposed to lung cancer; a model that has not undergone detailed scrutiny for appropriate end-point determination. We also sought to address previous inconsistency issues and maximise our pain detection capabilities. In our prior cancer investigations, mice underwent daily transfers to several different test enclosures with mice being lifted by their tail. However, Hurst et al. [5] showed adult mice find this type of lifting stressful, and that this stress can be substantially diminished by using non-aversive handling methods, such as cupping, or better, using a handling tunnel. These findings have now been widely replicated (How to pick up a mouse. Available online: https://www.nc3rs.org.uk/mouse-handling-research-papers, accessed on 22 December 2021). In addition to the welfare advantages of using non-aversive handling (NAH), the downstream impact of lowered anxiety may be to improve the quality of data those mice provide; both behaviourally and physiologically [6,7]. Handling-related stress could be especially problematic in mouse cancer investigations, not only because their welfare may already be compromised, but because anxiety and stress can either increase or reduce tumour growth rates [8,9,10,11]. It is consequently likely that our prior use of tail-handling and its high frequency contributed to the varied tumour growth rates we saw in our previous investigations, but moreover, it potentially increased variability in the other types of data that were collected [2]. In this study we, therefore, tried to limit the amount of handling needed for data collection by focusing on methods that can be applied to mice in their home-cages [12]. Thus, rather than using regular transfers to multiple types of test enclosures, we focused on home cage food and water consumption and nest building as metrics of daily well-being. Nest building was seen as a potentially highly useful welfare measure as it is a behaviour that mice are highly motivated to perform and any deterioration in ability can indicate compromised welfare in a wide variety of circumstances [13] but has so far not been used in mice with cancer. We still wished to determine if behaviour, mobility, and as seen in a previous cancer study, the onset of peripheral mechanical hyperalgesia [4] featured as potential indicators of declining welfare/pain as lung cancer developed. However, we limited these assessments (involving handling) to the beginning and end of the study. Likewise, the Mouse Grimace Scale (MGS) [14] was applied as a further potentially effective means of assessing whether mice are painful [15,16,17,18], but again, one that has so far not been used in cancer-bearing mice.

The study had several linked hypotheses; firstly, compared to controls, mice with lung cancer should show a decline in welfare potentially indicating escalating pain during tumour progression. If such were the case, because anxiety can escalate pain severity, we hypothesised that the welfare of mice undergoing NAH would be less detrimentally affected by cancer than those exposed to tail-handling. Finally, as a beneficial consequence of lowered anxiety, we hypothesised that mice undergoing NAH would provide more consistent data.

According to the results of daily welfare assessments (body weight, and water, but mainly food consumption), the mice experienced either pain or at least poor welfare beginning around 4 days before the study ended. Soon after this, there was a dramatic decline in nest building capability, suggesting these parameters could be used as a guide to end future similar studies as soon as these effects occur. Although this study did not find NAH improved welfare or data quality, neither did it show this technique had any negative impacts on welfare or the consistency of results.

## 2. Methods

All work was conducted at the Comparative Biology Centre, Newcastle University.

### 2.1. Animals and Housing

The study used 20 female and 20 male C57BL/6NCrl mice (Charles River Laboratories Inc, Margate, Kent, UK). At the beginning of the study (following the acclimation period) males weighed 25.4 ± 0.5 g and females 20.7 ± 0.2 g. They came certified free from common pathogens as listed on the Charles River website. Individuals were identified by tail marks using permanent non-toxic marker pens. Food (RM3 SDS Ltd., Waltham, UK) and water (Fresh20 water bags) were freely available. Mice were singly housed upon arrival in individually ventilated cages (IVCs—19 cm × 40 cm × 18 cm) (Arrowmight, Hereford, UK) with sawdust bedding (Gold Chip, BS and S Ltd., Edinburgh, UK). This was necessary to obtain individual welfare recordings. Each cage was provided with a 5 × 5 cm cotton nestlet (Datesand, Manchester, UK), a chew block and a cardboard tube (B&K universal, Hull, UK) for a 7-day acclimatization period prior to the start of the study. The animal room was maintained at 21 ± 1 °C, 48% humidity with a 12-h light cycle (lights on 07:00). Cages were cleaned weekly (between 08:00 and 10:00). The chew blocks and cardboard tunnels were renewed as necessary.

### 2.2. Animal Handling

On arrival, an equal number of male and female mice were randomly allocated (using an online random number generator, www.random.org, accessed on 22 November 2018) to two handling groups: standard tail-handling or a combination of the ‘cupping’ or ‘tunnel’ methods described by Hurst and West (2010). Once assigned, these were the only methods used when mice were handled. The only exception was that mice in the cancer group had to be restrained (scruffed) before being inserted into the device used for tumour inoculation (described below), however, the initial pick up from the home cage was carried out using the assigned handling method.

When a mouse assigned to the cupping/tunnel group had to be handled, if they were already inside the home-cage cardboard tube (tunnel), it was lifted in the tunnel and allowed to voluntarily exit onto a cupped hand. If they were not already in the tunnel, they were lifted using cupped hands. Tail-handled mice were always initially caught by the base of their tail, lifted, and briefly supported on the sleeve before undergoing the required procedure, e.g., weighing or transfer for other methods of data collection. Mice did not undergo any formal pre-study acclimation to the different handling methods because this would likely be impractical under normal circumstances and would, therefore, not reflect a ‘real word’ husbandry scenario.

### 2.3. Tumour Inoculation

Luciferase expressing tumour cells (Lewis Lung carcinoma: LL/2-luc-M38); were obtained from Caliper Life Sciences (Waltham, MA, USA) and were confirmed pathogen free by the IMPACT Profile I (PCR) at the University of Missouri Research Animal Diagnostic and Investigative Laboratory. Cells were cultured in Dulbecco’s Modified Eagle’s medium (DMEM) supplemented with 10% foetal bovine serum, 1% penicillin/streptomycin, and 0.1% sodium pyruvate (Invitrogen, Paisley, Scotland). On the morning mice were implanted, the cells were suspended in Dulbecco’s Phosphate Buffered Saline (DPBS) to a concentration of 1 × 10^7^ cells/mL. Ten male and 10 female mice were randomly selected (blocked by handling method), restrained in a Perspex tube (modified 50 mL syringe) and intravenously injected via the tail with 100 µL DPBS containing 1 × 10^6^ cells using an insulin syringe. They were then returned, via their pre-assigned handling method to their home-cages. The cages were transferred to a warming cabinet at 26 ± 2 °C for 1 h to allow the mice to recover. The remaining 20 mice were allocated to the control group. For welfare reasons, it was decided not to perform any control (DPBS only) injections. In the cancer group, one mouse died shortly after injection, and another four that failed to recover full mobility after 30 min were also euthanized. Overall, therefore, three males and two females were lost from the cancer group. How we attempted to balance the study is described in the data processing section.

### 2.4. Data Collection

All recordings or observations were made by treatment blinded scorers/observers (JVR or ALM).

Daily welfare assessment: baseline body weight and food and water consumption were recorded the day following tumour inoculation (Day 1). These data were then recorded every morning for the remainder of the study (8am–9am). Nest quality was also scored each morning when the mice were weighed, using an established protocol [19]. Mice were left on the balance in an appropriately sized container while this took place (<30 s). Once the nest was scored all nestlet material was removed from the cage and a new nestlet was placed onto the bedding, in the centre of the cage. The mice were then returned to their home-cages.

Behaviour assessment: baseline behaviour recordings were made on Day 1 between 9 and 12 pm, and then at the same time on the day each mouse was euthanized. Each mouse was removed from its home-cage and transferred to one of two clear plastic cages (32 cm × 16 cm × 13 cm; Techniplast UK Ltd., London, UK) containing only sawdust bedding. These cages were required by the analysis system to be clear of scratches, etc. so these were from stock that was only ever carefully hand washed. They were then filmed for 5 min using a video camera positioned approximately 20 cm from the cage front. Separate filming cages were used for males and females. An opaque divider was placed between the two cages to prevent mice from observing one another. Cages were cleaned between animals with 70% ethanol. These video recordings were then transferred to a PC and processed by HomeCageScan (HCS) software (CleverSystems Inc, Reston, VA, USA) for automated scoring of a range of mouse behaviours.

Nociceptive response assessment: On the morning of Day 2 batches of five same sex mice were transferred to an electronic von Frey (eVF) device to obtain a baseline nociceptive threshold recording using the ‘Mousemet’ eVF system (Topcat Metrology Ltd., Ely, UK; http://www.mousemet.co.uk, accessed on 22 December 2021). This apparatus had five raised clear plastic enclosures (19 cm × 4 cm × 10 cm) with steel rod floors and clear plastic lids and opaque dividers between enclosures. Males were followed by females until all mice had been recorded, cleaning all apparatus with alcohol each time. Because the mice were randomly assigned to the treatment group the individual mouse order was random. Once in their enclosure, they were allowed 5 min to acclimate. A nylon probe (0.5mm diameter at the tip) was used to apply a steadily rising pressure to the hind paw of each mouse; aiming for a force acceleration as close as possible to 1 gram per second (software guided). Once the foot was withdrawn, the system automatically recorded the peak force applied. Three measurements were taken from each hind paw with a minimum of 1 min between successive recordings. As far as possible recordings were made when the mice were not moving or performing other activities, e.g., grooming. The eVF recordings were repeated on the final day, as described below.

Imaging: Imaging began on the afternoon of Day 2 (12–2pm) with a baseline scan. Batches of three cancer-bearing mice were transported to a nearby room housing an IVIS Spectrum 200 bioluminescence imaging machine (PerkinElmer, Beaconsfield, UK). Each mouse was injected subcutaneously (s/c) with 150 mg/kg D-Luciferin (PerkinElmer, Seer Green, UK) diluted in DPBS, and then placed back into its home-cage. Ten minutes later the mice were anaesthetised in an induction chamber with 5% isoflurane in 2 L/min oxygen, and then placed back into the IVIS in dorsal recumbency on a stage heated to 36 °C. Anaesthesia was maintained by face-mask delivery of 2% isoflurane in 1.5–2 L/min oxygen. Subject depth was set to 1.5 cm and the exposure time to automatic. After a further 2 minute delay (a total of 12 minutes from injection of D-Luciferin) an open filter scan was made of the chest. The delay was to allow peak signal intensity to develop [3,4]. Mice were then returned to their home cages to recover, and once normally mobile, the cages were returned to the holding room. Imaging was repeated in the same way on the final day.

### 2.5. End-Point Determination and Terminal Data Collection

All mice were examined daily throughout the study during weighing and were euthanized when they reached end-point criteria. Several indicators were applied; a combination of which was used to judge the need for euthanasia. The process was akin to the methods often used to determine the need for the removal of mice from cancer studies.

The study ethical limits deemed that mice would undergo immediate cervical dislocation if they lost ≥20% of their body weight compared to their post-acclimation/baseline weight. Mice were designated as vulnerable by the NACWO or attending veterinarian (i.e., they were soon likely to reach end-point) if their weight had declined by more than 10%, or if there was between 5–10% weight loss accompanied by other signs of poor welfare, e.g., starey coat and/or poor mobility, or signs of laboured breathing. The nest of the relevant mouse was scored, and their consumption data was recorded. It then underwent 5 min of filming for HCS analysis and then underwent nociceptive threshold recording before and 1h after an injection of 20 mg/kg meloxicam (s/c). This was to determine if there was any benefit to the provision of analgesia at a late stage in disease progression; wherein any significant improvement would help us establish if what had been seen had been signs of pain and focus on these in future end-point predictions. Once these recordings were made the mouse had a final imaging session as the terminal procedure. It was killed by an overdose of isoflurane in the IVIS machine followed by cervical dislocation. Once the last tumour-bearing animal was removed from the study the controls underwent terminal recordings and were then killed by cervical dislocation.

### 2.6. Data Processing

Due to the post-inoculation deaths, three controls were transferred to the cancer group: one female and two males. This balanced the numbers by sex and handling method in the cancer group (*n* = 9) but meant the controls had one fewer male in the NAH group. Two further mice had to be excluded from the cancer group, reducing the group total to 16. This was because their tumours failed to engraft (determined from the imaging results).

The mean number of days before mice in the cancer groups reached the end-point was 21 ± 5 days and varied from 8 to 28 days. This meant that in addition to the euthanasia day there were 16 days of daily assessment data, (nest scores, body weight and food and water consumption) with enough cancer-bearing mice for comparison with those in the control groups (17 days in total). This is illustrated by the survival curve shown in Figure 1. These data, therefore, had to be evaluated in reverse, i.e., from 16 days prior to the day each mouse was euthanized (day 0 in Figure 2A–D). Data from mice in the cancer group were then paired with results from a procedure (handling method) and time-matched control. Put differently, for each cancer-bearing mouse, we paired data from a procedure-appropriate control recorded during the same study timeframe. For a more detailed description of this analysis approach see Roughan et al., 2014 [2]. This left a total of eight matched pairs for each treatment group. The final numbers that were entered into the statistical analysis are shown in Table 1.

The body weight and food and water consumption data were determined from the daily lab records, assessing absolute weights each day. Nest quality was scored from 1–5 [19]. The behaviour footage was transferred to a PC for analysis using HomeCageScan (HCS) software (Cleversys Inc., Reston, VA, USA). This system is hardcoded to recognise common mouse behaviours and provides an output of their frequency and duration (the list is shown here: http://cleversysinc.com/CleverSysInc/csi_products/homecagescan/, accessed on 22 December 2021). We have previously used this system in detecting the negative impacts of cancer in mice [2,3,4]. As in these previous studies we conducted exploratory analyses to determine which of the activities were most altered as cancer progressed, which were rearing, walking, inactivity, and grooming.

The HCS video footage was used to collect MGS data at baseline and on the last day. At least five images (‘screen grabs’) were obtained from each mouse and the images were then cropped (using Microsoft Picture Viewer) so only the face of the mouse was visible. Images were selected when, as far as possible, the mouse was not moving and was directly facing the camera. From these five, three images were randomly selected and placed into a custom designed excel scorebook, in random order. A study blinded scorer was given a description and a pictorial guide of the MGS [14] covering each of the five MGS Facial Action Units (FAUs); orbital tightening, nose bulge, cheek bulge, ear position and whisker position. Each FAU was scored on a 3-point scale (0 = not present, 1 = moderately present & 2 = obviously present). An average MGS score was then compiled for each mouse at baseline and on the final day.

The eVF thresholds were similar from the left and right paws of individuals, so all six readings were averaged. Living Image software (PerkinElmer, Beaconsfield, UK) was used to quantify the imaging data of lung tumour burden as the Total Flux (TF) of signals within 3 cm diameter circular regions of interest (ROI’s); one placed over the chest and another over the lower abdomen (to detect any metastases). We calculated maximum signal intensity for each mouse (on the final day) by subtracting the baseline signal intensity values from the respective chest or abdominal ROI’s; data are expressed as photons/sec/cm^2^. The rate of tumour development for each mouse was calculated as (TF(Final)-TF(baseline))/Total days enrolled.

### 2.7. Statistics

Individual mice were the experimental unit. The sample size was determined using nQuery Advisor (Version 6; Statsols, Cork, IRL). Previous results in mice with orthotopically implanted bladder tumours were found to lose weight alongside increased morphine-seeking compared to controls. Both outcomes provided strong evidence of a state of declining welfare, suffering, or pain. The present primary aim was to determine if the same effects occurred during the development of another internalised cancer (lung). Using bodyweight (often used as a gold standard welfare/end-point determinant) we used the previously found least significant group weight difference of 1.8 g and common standard deviation of 1.35 in a two-sided t-test (alpha set to 0.05). This indicated group sizes of 10 should have at least 80% power to detect the effects of tumour development. For the reasons given above the final group numbers were 8.

The daily welfare data were analysed using repeated measures General Linear Model (GLM) in SPSS software (version 23.0, SPSS Inc, Chicago, IL, USA). Data were tested for homogeneity of variance and sphericity, and when violated we used a probability correction factor based on adjusted degrees of freedom (Greenhouse-Geisser). Where this occurred, the corrected statistics are denoted by GG. For this analysis ‘Time’ (17 levels) was the within-subjects factor and ‘Treatment’ (Cancer or Control), ‘Handling’ (Tail or NAH) and ‘Sex’ were fixed (between-subjects) factors. Cage (individual) mouse was included as a random factor. Differences were considered statistically significant if *p* < 0.05. Pairwise (*Bonferroni* corrected) comparisons were used to identify individual days where significant treatment-related differences occurred.

The imaging data were used to estimate the average daily rate of lung cancer development. Image intensities on the day mice were euthanized were divided by the total number of days each mouse was enrolled, and the results were compared using the univariate GLM with the factors Sex’ and ‘Handling’. The MGS and behaviour data underwent a 2 × 2 × 2 repeated measures ANOVA with ‘Treatment’ ‘Handling’ and ‘Sex’ as between subject’s factors, and the eVF data underwent the same analysis with an additional level of ‘Time’ (i.e., baseline, and pre- and post-meloxicam). All values from the above analyses are expressed as means ±SEM.

The coefficient of variation was used to determine whether variability was lowered by NAH; calculated as the standard deviation divided by the mean and expressed as a percentage (i.e., (σ/$\bar{x}$) ×100), whereby higher values indicate increased data spread. CoV values from the period 16 to 7 days prior to euthanasia were determined for each daily welfare measure (i.e., body weight, food and water consumption and nest quality) and an average CoV was then calculated (pooled across the 4 parameters). The resulting set of values was then entered into a 2-way ANOVA with ‘Sex’ and ‘Handling’ method as between subject’s factors, the equivalent of Levene’s test for equality of between-groups variance.

CoV values from the last day behavioural recordings, from the MGS data and over the two final eVF assessments (pre- and post-meloxicam), were similarly compared, but for reasons given below, the impact of handling on these parameters were assessed in the cancer and control groups separately. Ignoring handling, the final day CoV values were compared within the cancer and control groups using ANOVA. Finally, the CoV of the average tumour growth rate was calculated to determine if handling method had any effect on tumour developmental consistency. All CoV values are expressed as percentages (±1 SEM).

## 3. Results

### 3.1. Survival Time

Figure 1 shows the percentage of male and female cancer-bearing mice remaining in the study as time progressed. The numbers of both sexes began to decline dramatically after 16 days. Three mice were euthanized based on weight loss alone (having reached the 20% weight loss end-point threshold). The Named Animal Care and Welfare Officer (NACWO) passed these three mice as fit to undergo terminal data collection (i.e., behaviour (+MGS), von Frey and then imaging).

### 3.2. Daily Welfare Assessment

The result of the daily welfare assessments is shown in Figure 2. Bodyweight showed a progressive decline in the cancer group, whereas the controls maintained or gained weight (significant ‘Time by ‘Treatment’ effect: (F(2.83, 67.987) = 14.061, *p* < 0.001 (GG)). Males and females lost similar weight and handling had no effect on weight changes as time progressed. The results of pairwise (*Bonferroni* corrected) comparisons are denoted by the asterisks in Figure 2A, indicating that the average weight of mice in the cancer group was lower than in the controls over the last 2 days.

Likewise, food and water consumption (Figure 2B,C respectively) also declined in cancer-bearing mice, whereas the controls ate and drank relatively more consistently. This was also during the last few days before euthanasia (significant ‘Time’ by ‘Treatment’ interaction: (F(6.718, 147.805) = 4.757, *p* < 0.001 (Food)); F(5.813,110.447) = 4.535, *p* < 0.001 (Water) (GG)). Pairwise (*Bonferroni* corrected) comparisons indicated days when the groups differed most significantly (asterisks in Figure 2B,C), showing food consumption appeared to be affected by tumour development earlier than water. Food and water consumption also were not affected by sex or handling method. Nest quality (Figure 2D) was largely the same in the cancer and control groups, but during the final 2 days, there was a more dramatic collapse in capability than in any of the other daily parameters, leading to a highly significant ‘Time’ by ‘Treatment’ interaction (F(8.504, 204.096) = 9.053, *p* < 0.001) (GG)). Nest quality was also unaffected by handling method or sex.

### 3.3. Behaviour (HCS)

As the study progressed the only activities that changed substantially related to the general ability of the mice to move. These were rearing, walking, and whether they were inactive or grooming. Although the HCS system produces data on both the frequency and duration of behaviour, the duration data were no more informative than the frequency results so only these data are described. Rearing (Figure 3A) showed a significant general decline over the course of the study (‘Time’ factor significant; (F(1,24) = 58.15, *p* < 0.001), and the change was far more obvious in mice with cancer (significant ‘Time’ by ‘Treatment’ interaction; F(1,24) = 11.71, *p* = 0.001). Rearing was not affected by sex or handling method. Figure 3B depicts several significant outcomes that were found in the analysis of walking behaviour. Tumour development caused walking to decline significantly over time (‘Time’ factor significant; F(1,24) = 105.1, *p* < 0.001), but again, more substantially in the cancer group (significant ‘Time’ x ‘Treatment’ interaction; F(1,24) = 22.54, *p* < 0.0001). Females were also generally more active than males (‘Sex’ factor significant; F(1,24) = 8.6, *p* = 0.007), and furthermore, females showed a significantly greater reduction in walking over the course of the study than males (significant ‘Time’ x ‘Sex’ interaction; F(1,24) = 13.96, *p* = 0.001).

As depicted in Figure 3C, the reduction in walking activity over time was modestly greater in females that had been tunnel handled (Significant 3-way interaction, ‘Time’ x ‘Sex’ x ‘Handling’ (F(1,24) = 4.79, *p*=0.038).

HCS was set to assign motionless periods to ‘sleeping’ if there was a total lack of movement in 95% of the preceding 5 s. Instances of this were only in the cancer group on the final day before the euthanasia group (on 53 ± 23 occasions). Grooming had similar tolerance settings, i.e., when minor movements were detected, but without mice travelling any distance. This was also only detected in the cancer group. There were 2.6 ± 1.8 grooming episodes at baseline and 42 ± 8.8 on the euthanasia day, resulting in a significant ‘Time’ effect (F(1,12) = 14.6, *p* = 0.002). Sleeping and grooming were similar in males and females and the frequency of each was unaffected by handling method.

### 3.4. MGS and Von Frey

MGS scores were similar between the cancer and control groups at baseline (0.87 ± 0.5 vs. 0.8 ± 0.45, cancer vs. control). They remained unchanged from baseline to euthanasia in control mice (0.77 ± 0.45) but were significantly increased in those with cancer (increasing to 2.3 ± 0.8, giving a significant ‘Time’ by ‘Treatment’; F(1,24) = 20, *p* < 0.001) (Figure 4A). There were no significant effects of handling and no effect of sex.

Compared to baseline, eVF thresholds had significantly decreased by the time of euthanasia (‘Time’ factor significant; F(2,48) = 6.1, *p* = 0.04; Figure 4B), but with no difference depending on handling method, and the decline was similar in both the cancer and control groups. Meloxicam had no significant analgesic effect, i.e., it did not reverse the apparent decrease in response threshold in either the cancer or control mice. Thresholds were lower in females both at baseline and at the two recording times on the day of euthanasia (‘Sex’ factor significant; F(1,24) = 21.8, *p* < 0.001) and the higher baseline thresholds meant males showed a relatively greater overall threshold reduction from baseline to the pre-meloxicam recording (‘Time’ by ‘Sex’ significant F(1,24) = 5.5, *p* = 0.026), but no difference thereafter.

### 3.5. Tumour Growth Rate

Tumour development was monitored via bioluminescent imaging. The two tail-handled male mice in which tumours failed to grow had average final signal intensities of 33 ± 2 × 10^3^ photons/sec/cm^2^/day. Given the final average signal intensity in the remaining tumour bearing mice was 3399 ± 856 × 10^3^ photons/sec/cm^2^/day, these two mice clearly failed to develop tumours. Excluding these two, signal intensity varied from 185 to 11,803 × 10^3^ photons/sec/cm^2^/day. This was still a highly variable rate and meant we could not detect significant differences according to sex or handling method.

### 3.6. Coefficient of Variation (CoV)

Average CoV values across the four daily welfare parameters were lower in mice in the NAH group but not significantly (NAH, 18 ± 1% vs. Tail, 21 ± 1%; *p* = 0.06). Values from the final day MGS, eVF and behaviour data were also not significantly different between the handling groups or sexes either in the control or cancer groups. Within the cancer group, the mice that had undergone NAH showed greater variability than those that were tail handled, but again not significantly (NAH, 91 ± 16% vs. Tail, 57 ± 14%). The equivalent values for mice in the control group were similar regardless of how they were handled (NAH, 46 ± 9% vs. Tail, 44 ± 7%). Sex had no effect on variability regardless of the parameter measured and there were no significant interactions. Overall tumour developmental variability was high in both handling groups, but highest in mice that were tail handled (NAH, 77% vs. Tail, 121%). Ignoring handling, a post-hoc ANOVA showed variability in the final day parameters was significantly higher in the cancer group (75 ± 11% vs. 45 ± 6%; ‘Treatment’ significant; F(1,49) = 5.1, *p* = 0.029).

## 4. Discussion

Mice used in cancer research are at significant risk of poor welfare due to the development of pain. Although there is no firm evidence, it is a generally accepted principle that reduced welfare is likely to reduce scientific data quality. In the case of cancer research applications it is therefore reasonable to argue that the upstream impact could be a negative impact on the translational validity of preclinical research findings. For example, if mice experience excess stress due to being handled inappropriately, or handled too often, this could be a nuisance factor that could have an important impact with regard to how they respond to cancer or to anti-cancer treatments. In this study, a refinement priority was to minimise the handling-related interference that potentially influenced the results of our previous investigations, where mice underwent multiple daily transfers to various types of test apparatus (e.g., for behaviour, von Frey and MGS recording). In this study, we only applied these types of recordings at the start and end of the study and assessed welfare on a daily basis using methods that could be applied while mice remained in their home-cages (i.e., food and water consumption, and nest building capability). Based on the eventual survival time of mice in the cancer group the analysis of daily welfare changes focused on the final 16 days prior to euthanasia. We aimed to establish the most appropriate end-points to be used for the LL2 model and to determine if the use of non-aversive handling (NAH or tunnel handling) impacted upon tumour development rate and the quality and consistency of all data collected. Bioluminescent imaging was used to try to accurately determine rates of tumour development and quantify metastases. This imaging method has previously been successful in mice with bladder tumours [3] and is currently one of only a few feasible options for longitudinal *in-vivo* monitoring of an internalised cancer, such as lung cancer.

Results showed that 3–4 days before euthanasia there was a clear reduction in all daily welfare parameters. Compared to controls, those bearing cancer showed reduced body weights over the final 3 days, which was most likely linked to reduced water, and in particular, the reduced food consumption that occurred between 4–5 days before euthanasia. Just before the mice were seen to require euthanasia, all cancer-bearing mice showed a dramatic reduction in nest building ability. There were also behaviour changes including a greater reduction in rearing and walking in the mice with cancer, whereas grooming and periods of total cessation of movement increased. These weight and behaviour changes were very similar to those we have previously observed in mice developing bladder or mammary cancer and were attributed to the occurrence of pain [2,4].

The article by Workman et al. [1] is one of the most widely cited reviews describing recommendations on the humane use of mice in cancer research. It suggests that body weight losses reaching 20% at any time, or a 15% loss maintained for >3 days would indicate a need for euthanasia, as would any substantial interference with normal locomotion or behaviour. Although in our study bodyweight losses averaged 7%, there were three mice that achieved 20%. However, none of these were assessed as being unfit to undergo terminal data recordings, so after undergoing behaviour and eVF recording they were anaesthetised for imaging and then euthanized. In these mice, and in all others that lost significant weight (>10%) there were accompanying reductions in our other behavioural indicators of welfare. Thus, it was highly likely that these mice suffered prior to euthanasia. Our findings, therefore, indicate a clear need to remove mice from similar studies several days earlier than present guidelines indicate.

Another of Workman’s recommendations is that researchers should apply all available knowledge to predict adverse effects in mice with cancer. The study of nest building as a welfare indicator was not common at the time of Workman’s publication. However, its assessment at the cage level is quick, simple, and objective [19], and it is known to be a crucial task for mice [20,21]. It probably derives from the need for a suitably warm environment to rear young and protect from predation. Here, we assessed the quality of the nests and showed mice will build new and complex nests, and these could easily be scored daily. As nest building is such an important task, we considered any alteration in this capability to indicate a significant welfare concern. As a basic ‘need’, it was not unexpected that the quality of nests in both the cancer and control groups was similar until the final 2 days; when build quality in the cancer group plummeted. Decreased nest quality is a vital welfare determinant in a variety of surgery and other disease models in mice [13], consequently, the substantial reduction in nest quality found here (approximately a 50% reduction in quality, 2 days prior to euthanasia) suggests this could provide an effective tool in end-point decision making in mice used in cancer studies. It was also notable that during the 2-week period prior to that decline, the cancer-bearing mice tended to construct superior nests. This may have been a by-product of tumour development; as tumours grow, blood flow is diverted towards them, reducing peripheral blood flow, and potentially resulting in an additional need for heat conservation.

There were no significantly different changes in eVF nociceptive thresholds from baseline to euthanasia between the cancer and control groups. Because thresholds declined significantly in both groups these data gave no indication that pain occurred. A simple interpretation of this is that it was due to sensitisation with repeated application of the eVF probe, but this was probably not the case given the two sets of recordings were temporally far apart. Another possibility is that it was the result of the additionally stressful events all mice experienced on the day they were euthanized. Whereas baseline eVF thresholds were the first data collected on day 2, on the final day the mice underwent behaviour recording, followed by two instances of eVF recording (before and after meloxicam), potentially worsening stress. The response to this can manifest as an endogenously mediated analgesic response known as stress-induced analgesia, resulting in increased nociceptive thresholds, but usually only in conditions of chronic or severely acute stress. On the other hand, less intense but nevertheless stressful circumstances may have an opposite short-term effect (i.e., hyperalgesia), which was most likely the case here [22,23].

A main aim was to determine whether NAH had any beneficial effects on welfare, as well as add to growing evidence that it can improve data quality and consistency [7,24,25]. We extracted CoV values from the daily assessment data from between 16 and 7 days before euthanasia, which allowed mice to have at least 16 days of prior differential handling. The cutoff of seven days was meant to avoid the potentially confounding effects of declining welfare in mice in the cancer group, and it was for this reason that the effect of handling method on the consistency of the end-stage recordings was assessed in the cancer and control groups separately. However, our assessment of variability found no convincing evidence of any beneficial effects on variability. Indeed, the only detectable effect of handling was a reduced impact on walking behaviour during the final day recordings, where, regardless of cancer status, females that had undergone NAH moved less than males. In line with the previously published findings on the effects of NAH, a possible explanation for this is that females that underwent NAH were to some extent ‘calmer’. Apart from that, there was no evidence of any positive effect on wellbeing since all mice in the cancer group developed signs of malaise regardless of their handling status. It is likely that the previously demonstrated beneficial effects of NAH were probably not substantial enough to impact positively on data quality or welfare in mice with cancer. However, it was still disappointing that NAH had no detectable benefit in the controls. Although this may indicate that no such result potentially existed, our methodology differed substantially from previous studies where, in terms of lowered anxiety, the beneficial effects of NAH were found to be robust and repeatable. The result in the controls may have been improved had we applied an exclusive period of NAH, typically for 1 week prior to the study, and used only handling tunnels rather than a mixture of tunnels and cupping. However, mice were only handled when it was unavoidable to mimic ‘real life’ husbandry, where there would not normally be time for this. All factors considered it was probably optimistic to expect NAH would result in significantly reduced variation or improved welfare. However, it is important to note that we found no escalation of variability due to NAH, which is a concern that researchers may use as a reason not to implement it [26]. It is further to be noted that there were also no signs of any increased harm to mice in the cancer group that underwent NAH.

The problem of variable tumour development impacted the method of data analysis; making it necessary to match data from cancer-bearing mice with controls from an equivalent elapsed study time. In a previous study of luciferin imaging in an orthotopic model of bladder cancer we also found a high degree of variability in our imaging results [3]. At that time, since the tumours had a similar post-mortem volume this was attributed to tumour necrosis/decreased vascularisation or reduced luciferin uptake rather than a variable burden. The physical burden of tumours that are primarily metastatic in their seeding characteristics (such as lung cancers) is more difficult to determine, even when undertaking histological analyses, which we did not. We were, therefore, unable to determine the aspect that ultimately resulted in such a high degree of variability in the imaging results. It may have been a result of differential luciferin absorption or poor injection methodology, but this was unlikely as the injections were given subcutaneously, and it has long been known that such errors are usually only a problem with the intraperitoneal route [27]. There were two mice that had very low final signal intensities (<1% of the total average). We can provide no viable explanation as to why these tumours failed to engraft but excluded these mice as a precaution. Data from the remaining tumour-bearing mice did not illustrate any significant effect of handling either in terms of mean growth rate or consistency.

In terms of husbandry, it would have been preferable to house the mice in groups. This is because they find single housing stressful, which could be another factor influencing (likely accelerating) tumour growth. However, all mice were treated this way, so it should not have influenced our findings. Single housing was essential to allow the collection of individual data on food and water consumption and nest building. As to the study design, another point worth mentioning is that at the time of tumour inoculation mice in the cancer group were scruffed and inserted into the inoculation device/tube, but the controls were not. This was thought sensible on welfare grounds, but with hindsight and because scruffing is stressful [24,25,28] we probably should have scruffed and restrained the controls also.

The post-inoculation mortality rate was particularly concerning, where one mouse died immediately, and four others had to be euthanized shortly afterward. These four mice appeared to have breathing difficulties, thus, cell clumping upon sequestration into lung tissue and venous occlusion may have been the problem, but there was no evidence of this upon post-mortem inspection. By necessity, the cells were cultured at a facility remote from the laboratory where they were used the same day, and this possibly resulted in some type of aggregation during transit, so perhaps this led to the unexpected deaths. The injections were performed by a very highly experienced technician, so the technique itself was unlikely to be the cause. Intravenous (orthotopic) tumour inoculation was the route recommended by the supplier as most likely to achieve maximum graft success, and therefore, provide a robust model of metastatic lung cancer, and in a clinical setting, more translationally relevant findings. Although in our case it was the route most likely to yield welfare concerns, in the future we would use the intra-tracheal route as a compromise to improve safety [29].

## 5. Conclusions

In conclusion, our findings strongly suggested that a welfare problem arose as lung cancer developed and that similar future studies would benefit from refinement, with mice being removed from the study at least 4 days earlier. However, the problem of highly varied tumour growth again made it difficult to determine this time-point accurately. This was probably also the reason that we could not detect any improvement in data consistency by implementing non-aversive handling. All factors considered, nest building and food consumption provided the most sensitive means of detecting the effects of cancer, so future similar studies could use these as general proxy measures of declining welfare.

## Figures and Tables

**Figure 1 animals-12-00023-f001:**
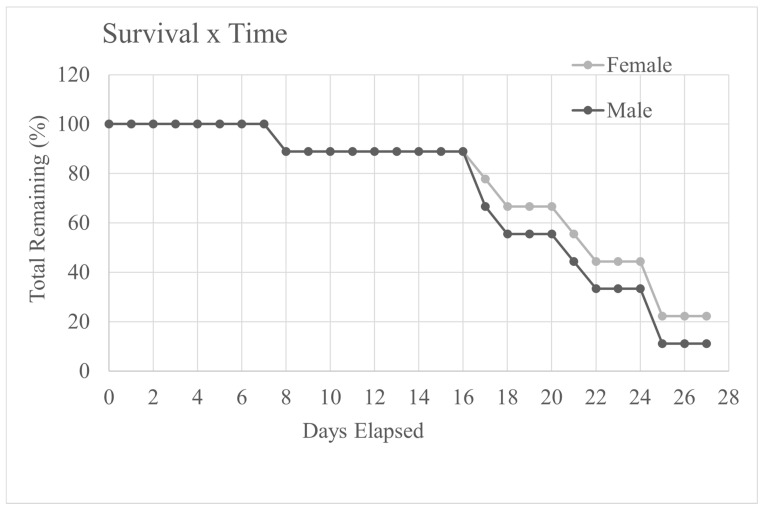
The percentage of male and female cancer-bearing mice surviving as the study progressed (excluding those that died immediately following tumour inoculation). Note the similarity of the decline in numbers of mice of both sexes beginning on day 16 (a total of 17 days following implantation on day 0).

**Figure 2 animals-12-00023-f002:**
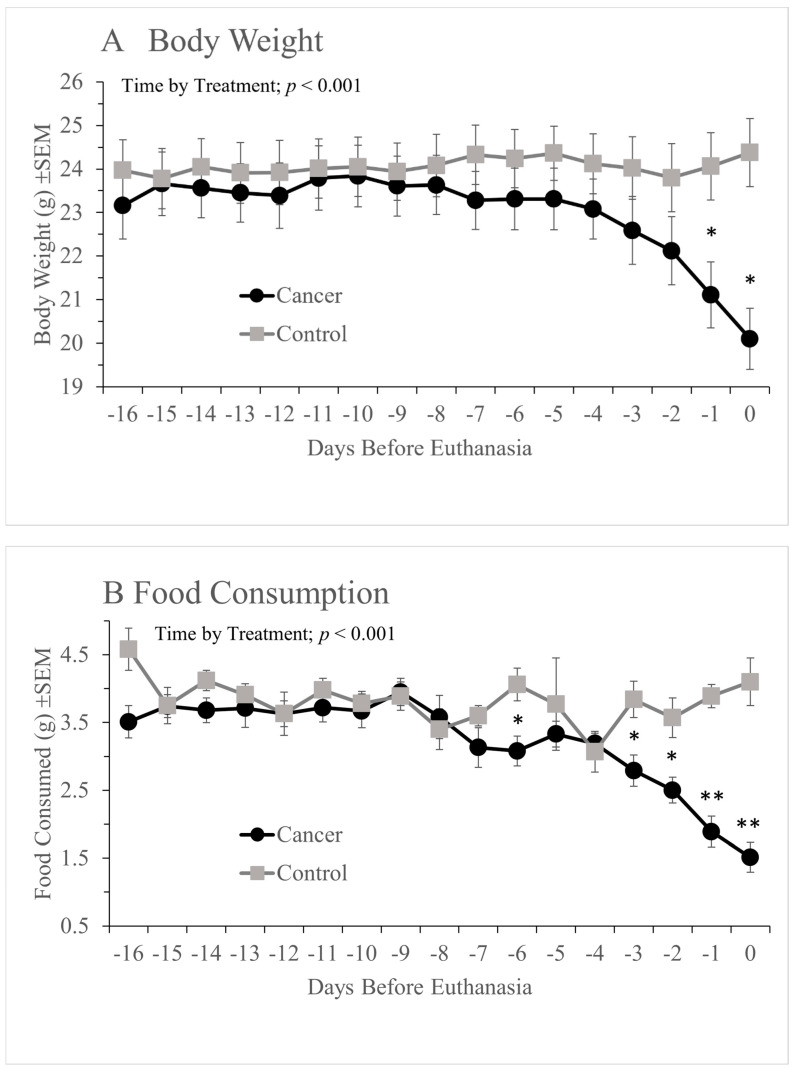
Mean daily body weight (**A**), food consumption (**B**), water consumption (**C**) and nest quality (**D**) in the cancer or control groups during the final 17 days before each mouse was euthanized (Day −16 to 0 inclusive). Asterisks denote the results of post-hoc analyses, i.e., days on which the cancer or control groups differed significantly (*, *p* < 0.05; **, *p* < 0.01: after *Bonferroni* correction). Note that while nest building (**C**) declined most dramatically (2 to 3 days before euthanasia), reduced food consumption was detected up to 4 to 5 days prior to euthanasia (**B**). Each figure (**A**–**D**) shows there was a significant ‘Time’ by ‘Treatment’ interaction; *p* < 0.001. Cancer/Control, *n* = 16.

**Figure 3 animals-12-00023-f003:**
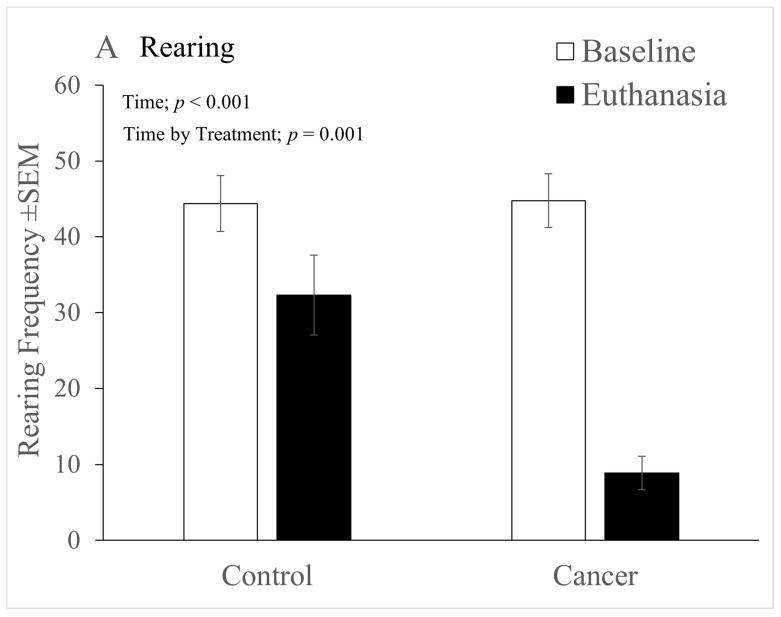
The mean frequencies of rearing (**A**) and walking behaviour (**B**, **C**) (±SEM) from baseline to euthanasia, indicating significant effects of the factors ‘Time’, ‘Treatment’, ‘Sex’ and Handling’ (with *p* values). (**A**) Rearing significantly declined in all mice over Time (‘Time’ factor significant), but more so in those with cancer (significant ‘Time’ by ‘Treatment’ interaction). (**B**) Regardless of the recording point females generally walked more than males (‘Sex’ factor significant), and all mice walked less as time progressed (‘Time’ factor significant). However, the decline in walking from baseline to euthanasia was relatively greater in females (significant ‘Time’ by ‘Sex’ interaction). Additionally, the overall reduction was greater in cancer bearing mice (significant ‘Time’ by ‘Treatment’ interaction). (**C**) Depicts that the greater decline in walking in females from baseline to euthanasia was primarily due to a greater reduction in walking movements made by those females that were in the NAH group (significant ‘Time’ by ‘Sex’ by ‘Handling’ (3-way interaction)). (**A**): Cancer/Control *n* = 16. (**B**): Cancer, male/female *n* = 7/9, Control male/female *n* = 7/9. (**C**): Tunnel, male/female *n* = 6/10, Tail, male/female *n* = 8/8. ♂: male; ♀: female.

**Figure 4 animals-12-00023-f004:**
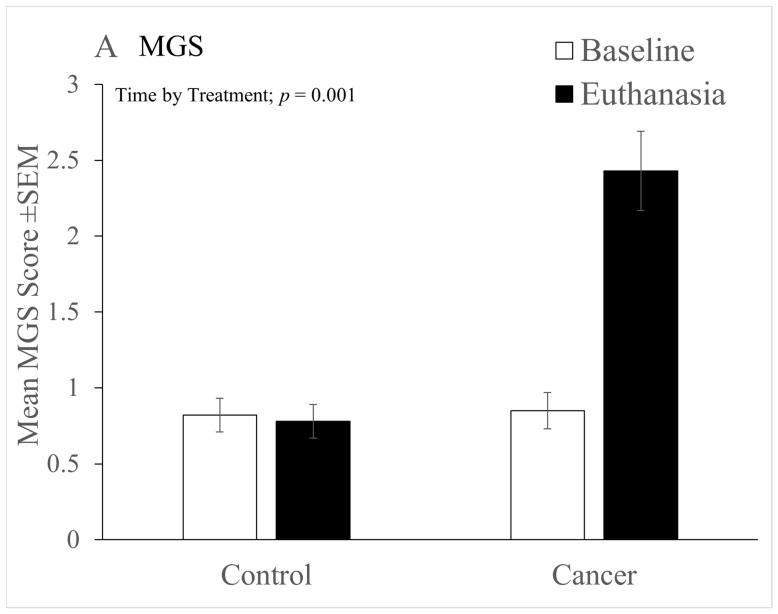
(**A**) Mean mouse grimace scale scores at baseline and on the day of euthanasia (±SEM) in cancer and control groups. Cancer-bearing mice showed a significantly greater increase in scores from baseline to euthanasia (significant ‘Time’ by ‘Treatment’ interaction). (**B**) Electronic von Frey (eVF) thresholds (recorded at baseline and on the day of euthanasia, when mice were recorded both before (Pre) and 1 h following (Post) administration of 20mg/kg meloxicam (M20). Note that males had significantly higher initial eVF thresholds so showed a relatively greater decline over the study duration (Significant effects of ‘Time’ and ‘Sex’ and significant ‘Time’ by ‘Sex’ interaction). Meloxicam treatment had no beneficial effect. Cancer/Control, *n* = 16 (seven males and nine females); Control, *n* = 16 (7 males and 9 females).

**Table 1 animals-12-00023-t001:** Final numbers of mice assigned to each treatment group in mice exposed to standard tail-handling or non-aversive handling (NAH).

	LL2 Cancer	Control
Tail-Handling	8 (4♂, 4♀)	8 (4♂, 4♀)
NAH	8 (3♂, 5♀)	8 (3♂, 5♀)
Total	16	16

♂: male; ♀: female.

## Data Availability

All data generated as part of this research are available through Newcastle University’s open data repository for researchers (Discover Research from Newcastle University. Available online: https://data.ncl.ac.uk/, accessed on 22 December 2021). Any other requests for assistance with accessing the source data should be made to the principal investigator, Johnny Roughan.

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
