# Peer review of "Welfare Assessment, End-Point Refinement and the Effects of Non-Aversive Handling in C57BL/6 Mice with Lewis Lung Cancer"

_animals, 2021, doi:10.3390/ani12010023_

Round 1

Reviewer 1 Report

The reviewed manuscript describes the effect of developing lung cancer in laboratory mice while a variety of welfare monitoring methods was applied. Tunnel handling showed no effect, whereas changes in general movement ability and facial expression could be observed. Most prominently nest building was impeded.

Comments:

The manuscript uses the term ›poor welfare‹ for the effects of carcinosis in a mouse model. But animal welfare is manmade, related to husbandry conditions controlled by humans. What the authors observe are typical symptoms of illness (reduction of movements, no nest building). They record experimental burden resulting from the pain of a cancer disease in the quest for a correct humane endpoint. The definition of poor welfare in connection with a disease model is questionable.

How could the progression of a disease connected with increasing suffering undermine data quality and translational validity? The change in behaviour is part of the experimental result, assumed that humane endpoints are applied to avoid too much suffering.

To recommend tunnel handling even if it does not help to evaluate the wellbeing of the mice makes no sense. Tunnel handling prevents a thorough health check during handling.

The authors claim that the final stage of the disease prevents their ability to recognize any beneficial effects of NAH. It means that they assume NAH has beneficial effects in this model and the progression of the disease impedes to observe them. How about, if there are no beneficial effects of NAH at all?! The study proves that NAH did not improve welfare or data quality.

Line 118: Temperature is not recognizable.

Line 152: Without a sham injection the effect of the i.v. injection itself is missing. Control mice were not restrained (but this is mentioned in the discussion). Animal welfare is a poor excuse for excluding an important control procedure.

Line 153: To lose 25% of i.v. injected mice implies a general problem with this injection method. We inject cancer cells via this route into sublethally irradiated immunodeficient mice since many years in our lab and never had such a loss rate.

Line 198: Only mice from the treatment group are submitted to anesthesia for in vivo imaging. Not mice from the control group. This might induce a difference in animal wellbeing or behaviour?!

Line 264, Figure 2: Food consumption decreases 2 days earlier compared to a reduced nest building. Therefore, Food consumption would be a better indicator for a humane endpoint compared to nest building!

Line 401, Figure 3: Figure 3A shows in the legend control and cancer and different columns for baseline and euthanasia. Figure 3B and 3C switches in the legend to baseline and euthanasia and shows columns with control and cancer instead.

Line 489: Reduced body weight was only measured one day before euthanasia according to Figure 2!

Line 501: To assess the animals fit to undergo terminal data recording was a subjective evaluation without any evaluation of suffering or pain (e.g. by stress hormone measurements).

Line 516: The ideal temperature for mice is under discussion. More recent papers of Gaskill and coworkers (2012, 2013a-d) showed in preference tests that mice prefer 20 centigrades when sufficient nest building material is offered.

Line 625, References: Some journal names are abbreviated, others not.

Author Response

REPLY TO REVIEWER 1: animals-1466031

REVIEWER: The manuscript uses the term ›poor welfare‹ for the effects of carcinosis in a mouse model. But animal welfare is manmade, related to husbandry conditions controlled by humans. What the authors observe are typical symptoms of illness (reduction of movements, no nest building). They record experimental burden resulting from the pain of a cancer disease in the quest for a correct humane endpoint. The definition of poor welfare in connection with a disease model is questionable.

AUTHOR: While the term welfare can be defined as you describe, it can also be used in the context of  “a measurable state in an animal which may be related to the adequacy of its ability to cope with its environment” [1]. We believe the measurements deployed in this study were directly applicable to assessing that coping ability. Food and water consumption and nest building became altered in a way that informed of a very high likelihood that the welfare of the animals was compromised. Conceptually therefore, we feel it is acceptable to refer to a state of ‘poor welfare’ with regard to our study.

REVIEWER: How could the progression of a disease connected with increasing suffering undermine data quality and translational validity? The change in behaviour is part of the experimental result, assumed that humane endpoints are applied to avoid too much suffering.

AUTHOR: Thank you for highlighting our lack of clarity on this point. So far as we know this is the first attempt to determine whether current end-point guidelines are appropriate for mice developing lung cancer. What we mean is, if mice are kept too long, then because they could begin to experience anxiety or pain, then findings related to the application of anti-cancer or other clinical interventions could become unreliable, undermining the translational validity of the study. It is already well established that excessive anxiety undermines data quality. We meant this would be relevant in future investigations where the objective is not necessarily an assessment of wellbeing related to endpoint refinement, which was the main objective in our study. We have modified the manuscript to make that objective clearer. See the first paragraph of the introduction.

REVIEWER: To recommend tunnel handling even if it does not help to evaluate the wellbeing of the mice makes no sense. Tunnel handling prevents a thorough health check during handling.

AUTHOR: We have not recommended the use of tunnel handling. As is normal, based on the many previously published reports of its beneficial effects, we hypothesised that it may have had a positive effect in a cancer research setting also. We included a web address where this evidence can be found: https://www.nc3rs.org.uk/mouse-handling-research-papers. In this case it did not, and we have discussed various possible reasons why. An important distinction that we make here is that most previous investigations of NAH used a pre-study acclimation period, when the mice usually experienced tunnel handling every day for 7-10 days. We wanted to make this as standard a replication of routine husbandry as possible, so no acclimation period was included. We also point out that we used a mixture of the 2 common NAH methods; not only tunnel handling but also cupping. We have added a statement to the discussion to clarify and discuss that NAH did not impact data quality or welfare which supports previous research on this topic.

REVIEWER: The authors claim that the final stage of the disease prevents their ability to recognize any beneficial effects of NAH. It means that they assume NAH has beneficial effects in this model and the progression of the disease impedes to observe them. How about, if there are no beneficial effects of NAH at all?! The study proves that NAH did not improve welfare or data quality.

AUTHOR: The additional discussion noted in the previous comment addresses this comment. Many institutions have now adopted this as the standard method of mouse handling (see for example https://www.nc3rs.org.uk/non-aversive-mouse-handling-practice and [2]) and we have tried to provide a plausible explanation for our result.

REVIEWER: Line 118: Temperature is not recognizable.

AUTHOR: Thank you – the text has been fixed.

REVIEWER: Line 152: Without a sham injection the effect of the i.v. injection itself is missing. Control mice were not restrained (but this is mentioned in the discussion). Animal welfare is a poor excuse for excluding an important control procedure.

AUTHOR: The response to the injection itself, if any, was not a topic of the investigation. Refining animal welfare is never a poor excuse for designing a study in such a way that it minimises the likely impact on the mice, especially when the effect of this is anticipated to be miniscule by comparison with the main factor of interest, i.e., the onset and development of cancer anticipated to begin several weeks later.

REVIEWER: Line 153: To lose 25% of i.v. injected mice implies a general problem with this injection method. We inject cancer cells via this route into sublethally irradiated immunodeficient mice since many years in our lab and never had such a loss rate.

AUTHOR: Yes, this was very concerning, and we admit this and offer a possible explanation. In the discussion we have expanded on the possibility that it may have been an issue caused in transit. The technician who injected the cells has more than 30 years of experience performing the technique and was just as surprised as us.. We have  stateed how we would try to circumvent this happening again.

REVIEWER: Line 198: Only mice from the treatment group are submitted to anesthesia for in vivo imaging. Not mice from the control group. This might induce a difference in animal wellbeing or behaviour?!

AUTHOR: We thought that previously also, which is why in a previous study all mice were imaged [3]. No difference was found between the cancer and control groups using the same Behaviour recognition software over the first 2 weeks. Again, on welfare grounds as well as for financial reasons it was decided to only image the mice with cancer because by the time they were anticipated to experience the effects of cancer (after several weeks) any impact of this was assumed would have become negligible.

REVIEWER: Line 264, Figure 2: Food consumption decreases 2 days earlier compared to a reduced nest building. Therefore, Food consumption would be a better indicator for a humane endpoint compared to nest building!

AUTHOR: Yes, that is a very good point. The combined knowledge gained from the consumption data along with the nest building data was the reason we advise to euthanase 4 days earlier. We highlighted the effect on nest building because it was the factor that declined most dramatically and is the easiest to measure at the cage side and provide an instant gauge of wellbeing. We have edited to text to reflect that food consumption was also effective (in the Abstract and Simple summary, in the Legend for Figure 2, as well as in the concluding part of the introduction, and in the discussion).

REVIEWER: Line 401, Figure 3: Figure 3A shows in the legend control and cancer and different columns for baseline and euthanasia. Figure 3B and 3C switches in the legend to baseline and euthanasia and shows columns with control and cancer instead.

AUTHOR: Yes, that is correct, but having re-read the results section describing these charts we can see where the confusion lies. The results section on the behaviour findings has been re-written. The chart labels have also been updated, as has the Figure 3 legend. We have replaced the Figure 3 charts with ones that include what statistics the charts refer to, and the significance levels. For consistency we have done the same in the other figures. Hopefully this makes things clearer.

REVIEWER: Line 489: Reduced body weight was only measured one day before euthanasia according to Figure 2!

AUTHOR: The legend states, the abscissa represents 17 days in total (day minus 16 to day zero). Therefore, including the measurement made on the morning of the euthanasia day, body weight was significantly reduced 2 days before euthanasia.

REVIEWER: Line 501: To assess the animals fit to undergo terminal data recording was a subjective evaluation without any evaluation of suffering or pain (e.g. by stress hormone measurements).

AUTHOR: Yes, it was. All forms of welfare evaluation have a subjective element. One decides on a measure, then interprets its meaning subjectively based on knowledge about the individual, what treatment or manipulations it has had, and the species and their natural history etc. It underlines the reason we have independent NACWO’s and veterinary staff to make decisions such as this. There was no option do assays such as for corticosterone as that would have further depleted the study numbers and added an excessive burden of stress.

REVIEWER: Line 516: The ideal temperature for mice is under discussion. More recent papers of Gaskill and coworkers (2012, 2013a-d) showed in preference tests that mice prefer 20 centigrades when sufficient nest building material is offered.

AUTHOR: Yes, this is a topic about which discussions will continue. In their 2012 article on the topic of heat preferences [4], Gaskill’s group reiterate that temperatures below 26 are below the preferred thermoneutral range of 26 to 32 degrees in mice; stating, “Overall, mice of different strains and sexes prefer temperatures between 26–29ºC and the shift from thermotaxis to nest building is seen between 6 and 10 g of material”. They maintain this stance in their 2017 article [5]. None of their studies conclude that mice preferred 20ºC if they had sufficient nesting material. The mice used in this study were housed well below the thermoneutral range (21±1), therefore it is very likely that they would have experienced some degree of cold stress, and why they were highly driven to build nest until the point when they were unable to .

REVIEWER: Line 625, References: Some journal names are abbreviated, others not.

AUTHOR: Thank you for highlighting this - Abbreviated journal titles have been added where necessary.

References:

  1. Brown MJ, Winnicker C. Chapter 39 - Animal Welfare. In: Fox JG, Anderson LC, Otto GM, Pritchett-Corning KR, Whary MT, editors. Laboratory Animal Medicine (Third Edition). Boston: Academic Press; 2015. p. 1653-72.
  2. J Y. News and Insights - JAX Blog [Internet]: The Jackson Laboratory. 2011. Available from: https://www.jax.org/news-and-insights/jax-blog/2011/june/minimizing-mouse-handling-stress-how-would-you-like-being-picked-up-by-your#.
  3. Lofgren J, Miller AL, Lee CCS, Bradshaw C, Flecknell P, Roughan JV. Analgesics promote welfare and sustain tumour growth in orthotopic 4T1 and B16 mouse cancer models. Laboratory Animals. 2017.
  4. Gaskill, B.N., Gordon, C.J., Pajor, E.A., Lucas, J.R., Davis, J.K., Garner, J.P., 2012. Heat or Insulation: Behavioral Titration of Mouse Preference for Warmth or Access to a Nest. PLOS ONE 7, e32799.. doi:10.1371/journal.pone.0032799.
  5. Johnson, J. S., Taylor, D. J., Green, A. R., & Gaskill, B. N. (2017). Effects of Nesting Material on Energy Homeostasis in BALB/cAnNCrl, C57BL/6NCrl, and Crl:CD1(ICR) Mice Housed at 20 °C. Journal of the American Association for Laboratory Animal Science : JAALAS56(3), 254–259.

Reviewer 2 Report

In this paper, the Authors evaluate experienced pain and/or poor welfare in mice developing lung cancer to establish if data quality and welfare can be enhanced using non-aversive handling.

The topic is very interesting and fit well within the scope of the journal but used methods were not carried out with scientific rigour and the results are not  of high-quality.

In addition, it is not reported where the observations took place and project authorisation number given by the competent authority. It is too restrictive to report only that "All procedures were conducted in accordance with the Animals (Scientific Procedures) Act 1986, European Directive EU 2010/63 and with the approval of the Newcastle University Animal Welfare and Ethical Review Body."

Furthermore, the conclusions were also presented in a reductive manner.

For these reasons, the manuscript is of poor quality and cannot be recommended for publication.

Author Response

REPLY TO REVIEWER 2: animals-1466031

REVIEWER: The topic is very interesting and fit well within the scope of the journal but used methods were not carried out with scientific rigour and the results are not of high-quality.

A: We are glad the reviewer found the article interesting and suitable for the theme of the special issue.

REVIEWER: In addition, it is not reported where the observations took place and project authorisation number given by the competent authority. It is too restrictive to report only that "All procedures were conducted in accordance with the Animals (Scientific Procedures) Act 1986, European Directive EU 2010/63 and with the approval of the Newcastle University Animal Welfare and Ethical Review Body.

AUTHOR: We have included a statement in the methods confirming that all work took place in the Comparative Biology Centre, Newcastle University, and added the license details under which the work was authorised.

REVIEWER: Furthermore, the conclusions were also presented in a reductive manner.

AUTHOR: We don’t know what ‘reductive’ means, please can you elaborate on this issue?

REVIEWER: For these reasons, the manuscript is of poor quality and cannot be recommended for publication.

AUTHOR: We do not feel that the above explanation of the problems the reviewer found in the manuscript are sufficient reasons to recommend rejection and would appreciate a more detailed evaluation of the manuscript.

Round 2

Reviewer 1 Report

Lines 23, 24: If appropriate endpoints are not applied, the animal experiment is illegal according to directive 2010/63/EU. That’s it. Data quality and translational validity are of inferior interest.

Lines 46-48: Some disease models are unfortunately connected with suffering of the animals (e.g. arthrosis models). The suffering can can only be reduced by analgetics. Such a general statement is not possible.

Lines 48, 49: Humane endpoints should prevent unnecessary suffering, not suffering at all.

Lines 71, 72: But only with adult mice and a small number of cages. Most of the studies used only a few mice (n=14,8,10,8,8,5,6-12,16,16,6,8). The only reference with pups have a lot of caveats (e.g. handling techniques were used infrequently, sanitization duration too short, total number of animals unclear, health, breeding and strain data missing). Use of clear acrylic tunnels (Hurst and West, 2010) deprives the mice of the possibility to hide. Method not suitable for jumpy or anxious mice or pups.

Line 110: Project authorisation number given by the competent authority is missing.

Line 117: Single housing applies a detrimental effect on the mice (Verwer et al. 2007; Van Loo et al. 2007; Madden et al. 2013). This increases the speed of tumor progression in the mice.

Line 226: Why have the mice to be earmarked (a painful method) when they are single housed? What means “possibly needing to be euthanized”? Do they need to be euthanized or not?

Figure 3: Figure 3A shows Baseline and Euthanasia on the x-axis and Control and Cancer as columns, whereas figure 3B shows it inverted. Figure 3C follows figure 3B. So figure 3A should be converted. Figure 4a and 4B is reciprocal again.

Lines 489-492: There is no evidence for this statement. Even this study cannot show it.

Line 536: The statement “research facilities are typically maintained at temperatures lower than mice prefer” is very disputable. Even the cited reference says: “… that it may not be possible to select a single preferred temperature for all mice.” Gordon et al. 1998 says: „Wood shaving bedding appears to provide an optimal range of thermal environment for mice… under standard housing conditions of 22°C.“ There are many detrimental effects of higher ambient temperature reported: Reduced pup weight (Valencak et al. 2013; Zhao et al. 2013), reduced mammary glands (Krol et al. 2003), reduced energy, fat and total solids in the milk, reduced  growth of sucklings (Krol and Speakman 2003), milk energy output and suckling time were lower (Zhao et al. 2016), decreased litter sizes and increased pup losse (Yamauchi et al. 1983), reductions in pup growth (Zhao et al. 2020), challenging for employees (Helppi et al. 2016; Gordon et al. 2017), increased sleeping apneas (Berteotti et al. 2020), increased male aggression (Greenberg 1972), reduced fertility (Helppi et al. 2016).

Lines 543-545: This is not very practical as indicator because it means to remove the nest every day and offer new nesting material to observe the nest building ability. To take the weight daily would be less disturbing.

Line 582: Typo: One the one …

Lines 593, 594: That there is no increased harm in the tunnel handling group is not really an argument to use this method. Then everything else which induces no additional harm could be recommendable.

Author Response

Dear reviewer, thank you for taking the time to reconsider our article. We have amended the manuscript in line with the various helpful suggestions. Below you will find the replies to specific remarks on a point-by-point basis. Thanks again.

REVIEWER: Lines 23, 24: If appropriate endpoints are not applied, the animal experiment is illegal according to directive 2010/63/EU. That’s it. Data quality and translational validity are of inferior interest.

AUTHOR: Very good point. Text deleted.

REVIEWER: Lines 46-48: Some disease models are unfortunately connected with suffering of the animals (e.g. arthrosis models). The suffering can can only be reduced by analgetics. Such a general statement is not possible.

AUTHOR: Yes, another good point, and we also have models where some unavoidable suffering is allowed. Text amended in line with this. We now say: “If mice suffer unnecessarily or beyond the limits that a particular study allows, then they may not yield suitably reliable findings”.

REVIEWER: Lines 48, 49: Humane endpoints should prevent unnecessary suffering, not suffering at all.

AUTHOR: Agreed, the word ‘unnecessary’ has been added before ‘suffering’.

REVIEWER: Lines 71, 72: But only with adult mice and a small number of cages. Most of the studies used only a few mice (n=14,8,10,8,8,5,6-12,16,16,6,8). The only reference with pups have a lot of caveats (e.g. handling techniques were used infrequently, sanitization duration too short, total number of animals unclear, health, breeding and strain data missing). Use of clear acrylic tunnels (Hurst and West, 2010) deprives the mice of the possibility to hide. Method not suitable for jumpy or anxious mice or pups.

AUTHOR: Yes, there are many circumstances that could limit the effectiveness of tunnel handling. The numbers of mice or cages previous studies used is irrelevant so long as they were designed appropriately and had sufficient power. It was up to the institutional and scientific reviewers to challenge the numbers used in those studies. It is probably the robustness of the effect that has allowed low group numbers to achieve significant findings, and this was what encouraged us to try it. Whether it is appropriate for pups has no relevance to our study. According to Hurst, depending on the mouse strain, the technique can take weeks for a beneficial effect to be detected, and therefore we discussed this as a potential limitation. Yes, it probably is more difficult in more jumpy strains, but Hurst et al say in the FAQ section of the NC3Rs website they have managed to use tunnels in wild house mice, wood mice, harvest mice, bank voles, field voles, laboratory rats and fat dormice. As they say, “Indeed, every small rodent that we have tried”. We cannot comment on how successful they were as no data have been published so far. The FAQ by Hurst also mentions that there is no need to remove other forms of enrichment such as shelters/hiding places, or you can use an opaque tunnel. We felt that because the mice had sufficient nesting material that his was suitable for hiding. To respond to this all we can do is add the caveat ‘adult’ to the lines in question, which we have done.

REVIEWER: Line 110: Project authorisation number given by the competent authority is missing.

AUTHOR: AWERB approval number has been added to the Institutional Review Board Statement.

REVIEWER: Line 117: Single housing applies a detrimental effect on the mice (Verwer et al. 2007; Van Loo et al. 2007; Madden et al. 2013). This increases the speed of tumor progression in the mice.

AUTHOR: Yes, being a stressor in mice it probably does, but given all mice experienced this, it should not have had any bearing on our results. We have expanded on this at the relevant point in the discussion (lines 620-624 in the version with tracked changes).

REVIEWER: Line 226: Why have the mice to be earmarked (a painful method) when they are single housed? What means “possibly needing to be euthanized”? Do they need to be euthanized or not?

AUTHOR: Sorry, we did not mean this literally; it was just a term meaning they were designated as vulnerable and likely to achieve end-point criteria very soon. These decisions are difficult, and we sought appropriate advice as to whether they needed to be euthanased immediately or not. None underwent any recordings if they were already at endpoint of ≥20% weight loss. We have tried to clarify the decision-making process including the term ‘designated’ rather than earmarked.

REVIEWER: Figure 3: Figure 3A shows Baseline and Euthanasia on the x-axis and Control and Cancer as columns, whereas figure 3B shows it inverted. Figure 3C follows figure 3B. So figure 3A should be converted. Figure 4a and 4B is reciprocal again.

AUTHOR: Thanks for the suggestion. Figure 3A have been converted to be consistent with 3B and C. Figure 4A has also been converted as the reviewer suggests, so the grouping shown on the x-axis and the 2 series illustrates the time the recordings were made (Baseline vs Euthanasia). We did the same for figure 4A (MGS). Similarly, we converted figure 4B (eVF) so that the male and female groups were represented on the x-axis grouped by time. However, this chart has no cancer and control group differentiation as the groups were similar, and the sex factor was the most important thing to illustrate. The converted version made this less clear in our opinion so we feel figure 4B should stay as is.

REVIEWER: Lines 489-492: There is no evidence for this statement. Even this study cannot show it.

AUTHOR: Yes, its true we have not demonstrated this, but it is still a widely accepted general principle and one that is clearly laid out stated in the NIH guidance document on Recognition and Alleviation of Pain in Laboratory Animals. See Chapter 4: Consequences of unalleviated pain (https://www.ncbi.nlm.nih.gov/books/NBK32658/). The first paragraph of the discussion has been modified to explain this stance, and why we designed the investigation to limit the effect of potential nuisance factors such as stress from excessive handling.

REVIEWER: Line 536: The statement “research facilities are typically maintained at temperatures lower than mice prefer” is very disputable. Even the cited reference says: “… that it may not be possible to select a single preferred temperature for all mice.” Gordon et al. 1998 says: „Wood shaving bedding appears to provide an optimal range of thermal environment for mice… under standard housing conditions of 22°C.“ There are many detrimental effects of higher ambient temperature reported: Reduced pup weight (Valencak et al. 2013; Zhao et al. 2013), reduced mammary glands (Krol et al. 2003), reduced energy, fat and total solids in the milk, reduced  growth of sucklings (Krol and Speakman 2003), milk energy output and suckling time were lower (Zhao et al. 2016), decreased litter sizes and increased pup losse (Yamauchi et al. 1983), reductions in pup growth (Zhao et al. 2020), challenging for employees (Helppi et al. 2016; Gordon et al. 2017), increased sleeping apneas (Berteotti et al. 2020), increased male aggression (Greenberg 1972), reduced fertility (Helppi et al. 2016).

AUTHOR: OK, as this is such a complex and controversial issue, we have removed the suggestion and citation.

REVIEWER: Lines 543-545: This is not very practical as indicator because it means to remove the nest every day and offer new nesting material to observe the nest building ability. To take the weight daily would be less disturbing.

AUTHOR: Nest scoring is no less disturbing or practical than daily weighting because both require opening the cage and removing the mouse. If you do not renew the nest there is nothing to record the next day. To clarify, the nesting material was replaced each day, and the mouse had 24 hours to construct a new nest, and this was then scored the next day. Practically speaking it’s the same with weighing every 24h. The reviewer may be thinking about the time-to-integrate-to-nest test (TINT), whereby mice are offered a portion of new material and the time they take to incorporate it into their existing nest is measured (usually within 10 minutes)

REVIEWER: Line 582: Typo: One the one …

AUTHOR: Thanks for spotting this. It has been corrected.

REVIEWER: Lines 593, 594: That there is no increased harm in the tunnel handling group is not really an argument to use this method. Then everything else which induces no additional harm could be recommendable.

AUTHOR: We are not saying anything non harmful is recommendable. Tunnel handling was tested in response to a survey we undertook on the reasons why scientists (among other stakeholders) seemed to be reluctant to use it. One of the main reasons given was a concern that it would harm data consistency (1). It was because of this that we undertook the reliability assessment. This is still an important point to make and does not mean we are recommending everything that is not harmful should be used.

(1) Henderson LJ, Smulders TV, Roughan JV. Identifying obstacles preventing the uptake of tunnel handling methods for laboratory mice: An international thematic survey. PLoS One. 2020;15(4):e0231454.

Reviewer 2 Report

Requested changes were performed and made revisions have considerably improved your manuscript. It is my opinion that the paper is adequate for publication.

Author Response

Dear Reviewer, thank you for taking the time to reconsider our manuscript. We have made some further minor changes to improve clarity.